# Spatio-Temporal Modeling and Simulation of Asian Soybean Rust Based on Fuzzy System

**DOI:** 10.3390/s22020668

**Published:** 2022-01-16

**Authors:** Nayara Longo Sartor Zagui, André Krindges, Anna Diva Plasencia Lotufo, Carlos Roberto Minussi

**Affiliations:** Electrical Engineering Department, UNESP-São Paulo State University, Av. Brasil 56, Ilha Solteira (SP), Sao Paulo 15385-000, Brazil; nayara.sartor@ifmt.edu.br (N.L.S.Z.); krindges@ufmt.br (A.K.); anna.lotufo@unesp.br (A.D.P.L.)

**Keywords:** advection-diffusion problem, modeling and simulation Asian soybean rust, fuzzy logic

## Abstract

Mato Grosso, Brazil, is the largest soy producer in the country. Asian Soy Rust is a disease that has already caused a lot of damage to Brazilian agribusiness. The plant matures prematurely, hindering the filling of the pod, drastically reducing productivity. It is caused by the *Phakopsora pachyrhizi* fungus. For a plant disease to establish itself, the presence of a pathogen, a susceptible plant, and favorable environmental conditions are necessary. This research developed a fuzzy system gathering these three variables as inputs, having as an output the vulnerability of the region to the disease. The presence of the pathogen was measured using a diffusion-advection equation appropriate to the problem. Some coefficients were based on the literature, others were measured by a fuzzy system and others were obtained by real data. From the mapping of producing properties, the locations where there are susceptible plants were established. And the favorable environmental conditions were also obtained from a fuzzy system, whose inputs were temperature and leaf wetness. Data provided by IBGE, INMET, and Antirust Consortium were used to fuel the model, and all treatments, tests, and simulations were carried out within the Matlab^®^ environment. Although Asian Soybean Rust was the chosen disease here, the model was general in nature, so could be reproduced for any disease of plants with the same profile.

## 1. Introduction

The state of Mato Grosso is one of the largest grain producers in Brazil. In this way, its economy depends significantly on the success of the harvest of its crops. This, in turn, depends on variables determined by nature, such as rain and/or sun at the right times, while others that can be controlled, even partially, by man. These are the cases of diseases in plantations, which can be as decisive as rain and sun.

Given the importance of containing these pathogens, numerous studies are conducted frequently. Understanding what causes them, how they are disseminated, identifying their symptoms, among others, is crucial to determining how to avoid and/or control them. Thus, researching their evolution is of great value to understanding the whole process.

When it comes to growing grains, it is impossible not to consider soy. Similarly, when it comes to the diseases that affect it, it is impossible not to remember the Asian Soybean Rust (ASR). When it is not controlled, it causes serious economic damage to the farmer.

There are no barriers to the *Phakopsora pachyrhizi* (PP) causative of this disease. Their spores are carried by the wind, and when they reach a host, with a few hours of favorable conditions, they can germinate and subsequently generate more spores, which are also carried away by the wind, and so on. Its main host is the soybean plant, but in the off-season, it can survive on alternate hosts. Thus, if it is not contained, the disease gains unimaginable proportions.

It is in this sense that partial differential equations (PDE) and fuzzy logic (FL) [1] arise, important mathematical tools used in the modeling of numerous problems of the most varied areas. They are able to systematize the phenomena of nature, economic and social behaviors, among others, often enabling the prediction of their developments at certain times, or under certain circumstances.

Therefore, this research aims to understand the evolution of ASR, as it is a disease of great economic importance for the state of Mato Grosso, and also to estimate it. By predicting its occurrence with acceptable accuracy, it is possible to reduce fungicide applications, which generates both monetary savings and saves the environment from the damage that these applications cause. Thus, the spore concentration will be estimated by means of PDE, the disease-friendly environment by means of FL, and the relationship between these variables, and also between the existences of hosts, will be developed by means of FL. Thus, the problem will be modeled by combining these tools. The motivation of this architecture is to produce an intelligent system that is able to guide the decision-making of farmers.

## 2. Main Related Works

Soy is a grain that, in one way or another, is present in the preparation of food of many families. Soybean oil, soy milk, and soy meat are just a few examples of groceries produced from it. But its importance is not limited to this. Its bran is also used to manufacture feed to feed the most diverse herds, and with its oil, it is possible to produce energy. Its cultivation in Brazil takes place throughout the entire territory where there is agriculture. Thus, it has become, in recent decades, one of the most exported products by the country.

Its scientific name is *Glycine max*, and it is part of the legume family, as are peas and beans. It is an upright, thick and annual herbaceous species that can reach up to 1.5 m in height. It has a multitude of uses in the food and industrial sectors and represents one of the main sources of edible vegetable oil and proteins for the use of animal feed [2].

Soybeans are considered one of the oldest crops, having emerged in China millennia ago [3,4]. It came to the West in the 20th century, when the United States began its cultivation and, in a few decades, the growth of the cultivated area became exponential. In Brazil, its commercial exploitation began to gain importance from the 1940s, in Rio Grande do Sul. Its production grew significantly in the 1960s and established itself as the main product of Brazilian Agribusiness in the 1970s, in the southern region of the country. Between the 1980s and 1990s, the cultivation of the grain expanded significantly also to the tropical region of Brazil [4].

In the 2012/13 harvest, 267.48 million tons of soybeans were produced in a cultivated area of 102.93 million hectares. The United States, Brazil, and Argentina are the largest producers of this oilseed, being responsible, together, for the production of about 80% of the soybean consumed in the world [3].

The expansion of the cultivated area, however, mainly runs into problems with pests, diseases, weeds, etc. Many pathogens, such as fungi, bacteria, viruses, and nematodes, attack soy. However, the disease caused by the PP, has been considered one of the most important. It was first reported in Japan in the mid-1900s and, by 1914, had already spread to several Southeast Asian countries [5].

In Brazil, it was identified in 2001, and in the 2002/03 crop year, it had already spread throughout the main producing regions of the country, causing significant losses. In the 2003/04 crop, adding expenses with acquisitions and fungicide spraying to losses caused by reduced yields, the estimated embezzlement exceeded the amount of two billion dollars [6].

The incidence of ASR varies from one region to another, depending on the water regime, off-season climatic conditions, and crop management [7]. In any case, it is a disease of difficult prediction. Its initial occurrence and severity depend on the climatic conditions and the proximity of the inoculum source, which can vary greatly from one year to another. In addition, the different climatic conditions and the existence of voluntary plants (in crops or on the banks of roads) or susceptible host plants make it impossible to make a recipe that meets all soybean-producing regions [8]. Thus, understanding the disease according to spatial and time aspects allows the obtaining of useful information that enables the construction of an efficient and rational control program [7].

Given all the above, the need to understand the ASR is evident. In general, all the complexity that involves the disease is what motivates researchers from all areas, in the search for remediation or at least prevention. Thus, several studies are carried out with the intention of controlling it, of classifying the risks of its occurrence, of determining its favorability, its severity, in short: with the intention of predicting it.

The invasion of ASR revealed how vulnerable agricultural ecosystems are to pathogens of airborne plants. And it is this vulnerability that has aroused broad interests in the development of disease predictions [9]. Table 1 presents in a systematized way, some research carried out with the intention of controlling the ASR, of classifying the risks of its occurrence, of determining its favorability, and its severity, in short: with the intention of predicting it. In addition, also listed is research that associates artificial intelligence and agricultural monitoring.

From the mentioned works, it is possible to perceive the use of artificial intelligence, differential equations, or the combination of both in the development of the proposed models. In general, everyone intends to contribute to the efficiency of agricultural systems and/or, where possible, to foresee the behavior of the ASR. This is also the objective of this study, and for this purposes, the fuzzy logic techniques will be combined with those of the Diffusion-Advection Equation in the development of the model.

## 3. Proposed Methodology: Modeling of The Space-Temporal Dynamics of Asian Soybean Rust

The objective of this research was to model the temporal and spatial development of ASR in a given region. For that, a fuzzy system was developed that relates favorable environmental conditions for disease progress (favorability), the concentration of the PP (spores), and the existence of host plants (soybean).

FL can be seen as one of the most effective mathematical tools for dealing with the problem of ignorance and subjectivity, inaccuracies, and partial truths, enabling the handling of real-world problems, often with low-cost solutions [21]. Thus, to determine how favorable the climatic conditions are for the disease to develop, the Fuzzy System (FS) developed by [14] was used.

To estimate the concentration of the pathogen, a diffusion-advection equation (DAE) was constructed capable of modeling its dispersion. The DAE is widely used to model the most varied phenomena. In this way, it is well known in the scientific environment.

For the presence of host plants, soybean-producing regions and their surroundings are considered. In addition, localities near highways are also considered, since the voluntary soybean plants that multiply along the roads also favor the conservation of the fungus.

The region chosen for the development of the research was the territory limited by the state of Mato Grosso, in the Midwest region of Brazil. It is the largest soybean producer in the country. In the 2020/2021 harvest, alone, it was responsible for producing 26.5% of all Brazilian soybeans, which corresponds to almost 36 million tons [22].

Thus, maps of favorability, spore concentration of the fungus, and the presence of host plants were constructed. These maps constitute the values for the input variables of the FS that estimate the chance of a region being affected by Asian soybean rust. This estimate will be given the fuzzy occurrence name. For the records of actual occurrences, those made available by the Antirust Consortium were used, between the 2015/2016 and 2019/2020 harvests.

In the construction of the map that indicates environmental conditions favorable to disease progress, data were considered between the end of 2018 and the end of 2019 of the National Institute of Meteorology (INMET). They were obtained from collections carried out by meteorological stations of automatic surface, located in the state and in neighboring regions. The same was done in relation to the construction of the vector field that constituted the DAE for the spore concentration map. In the Matlab^®^ environment, the input data, the generation of values for all maps as well as their images, were treated.

For the development of these maps, a mesh was constructed with free software Gmsh, from the coordinates of the state contour. The mesh is formed by 7404 border points and 2,704,911 internal points. The coordinate data are provided by the Brazilian Institute of Geography and Statistics (IBGE). Thus, it was possible to assign to each point of the mesh, proximity value with hosts, spore concentration, and favorability of environment.

### 3.1. ASR Favorability Map

ASR favorability maps have been built for soybean-producing states for some years. In their research, [23] reported the generation of monthly, and weekly favorability maps for the state of Paraná. Data were obtained from information from weather stations. The maps were part of the basis of the epidemic warning system for the disease, developed by the ABC Foundation with the support of universities, institutes, and companies in the agricultural sector.

The development of the ASR Favorability map for the state of Mato Grosso was based on the figure created by [14], which presents the areas favorable to the development of ASR in the state of Minas Gerais. The authors estimated them by means of an FS, which was fed with monthly average January temperature observations from 39 INMET weather stations for the period 1961 to 1990, and with a leaf wetness period fixed at 12 h, using altitude, latitude, and longitude as covariates. Thus, data available for the state of Mato Grosso were collected and treated for use in the FS. Then, the obtained values were interpolated, enabling the construction of the map.

To feed the FS, records from 50-INMET meteorological stations based on sensors, and measurement systems were used, located in the state and in neighboring regions, were used. Data is available hourly for the last 365 days. Due to this limitation, data recorded between 21 December 2018, and 20 December 2019 was considered.

In the automatic stations, for the input variable Temperature (°C), it is recommended that the daily value be obtained from the arithmetic mean of all records for the day [24]. Instant, maximum and minimum temperature data are available, for each hour of the day. Thus, all values for the day were summed up and divided by the number of observations. Then, a 365-day matrix by 50-stations was built for this variable.

As for leaf wetness, a simple method for its determination is to consider it equal to the number of hours with a relative humidity of the air above a pre-established value. In general, 85% (Number of Hours of Relative Humidity—NHRH ≥ 85%), 90% (NHRH ≥ 90%) or 95% (NHRH ≥ 95%) are taken as a limit value [24]. Thus, for the Leaf Wetness (h) input variable, relative humidity of air data was used. For these records, instant, maximum, and minimum data are also available. Thus, the arithmetic mean between these values per hour was made, which generated up to 24 relative humidity data per day per season. These data were interpolated, generating a value for every minute of the day. Leaf wetness was then determined by the sum of the minutes that had RH ≥ 90%, converted into hours, and stored in a 365-day matrix.

As can be seen, the processing of the data of entry of the FS for the elaboration of the map of Mato Grosso is different from that given to the data of Minas Gerais. This is due to the fact that in the region of this research there are few conventional meteorological stations, which made it impossible to elaborate measures from a historical series. On the other hand, the records made for all hours of the day in the automatic stations allowed the daily estimation of leaf wetness. Thus, it was not necessary to keep it constant in 12 h, as the authors did, in the elaboration of the map for Minas Gerais.

For the interpolation, the inverse distance weighting (IDW) and Kriging models were tested. The IDW, or inverse of distance, is the most used local deterministic model, and which produces a result similar to that of ordinary Kriging. As for Kriging, it is a geostatistical technique for estimating values of variables distributed in space or time, or in both, based on close values when considered mutually dependent by variographic analysis [25].

### 3.2. PP Spore Concentration Map

To estimate the spore concentration of the PP, which causes ASR, a DAE was constructed, which is composed of source, transport, diffusive term, and decay. These are, respectively, places where the pathogen is “produced”, the vector field that takes it from one place to another of the system, its own dispersal capacity, and its mortality rate.

Sources were located along all major highways in the state. Reference [26] explains that because it is a biotrophic pathogen, the agent responsible for the disease only multiplies, and survives in living tissues. Thus, voluntary soybean and alternative hosts ensure the survival of the fungus, giving rise to the initial inoculum for the next epidemics. In addition, whenever soybeans are harvested and transported, the grains that fall along the roads germinate in the first rains, giving rise to plants that can be infected by the pathogen. Thus, they favor their survival during the off-season, serving as a link between one crop and another, or a green bridge [27].

The problem of voluntary plants existing next to the highways is more serious in the states of Mato Grosso, Minas Gerais, and São Paulo, where the concentration of central pivot areas is large [28]. The grains that fall on the roads of Mato Grosso, during the flow of the harvest, have caused considerable damage with the proliferation of diseases. Thus, the rust spores are spread by the highways of the Central-West region of Brazil [29].

The areas of land situated on the side of state highways are of public interest and correspond to the recoil range. They are 20 m to the right and 20 m to the left, measured from the central axis of the highway [30]. Therefore, slightly increasing this limit, every node of the mesh that is far up to 50 m from some main highway was used as a source.

The transport was represented by the vector field obtained from the data of wind speed and direction, available in the INMET automatic stations. In this way, at all points of the mesh, for all days of the year, it was possible to obtain a corresponding vector. Thus, the transport was variable each day, from element to element.

In addition, vectors parallel to the roads were added to the vector field at all nodes of the grid that are up to 1 km away from any main road. The dimensions of these vectors are proportional to the distance measured from the highway (the closer, the greater the intensity of the vector). Thus, it was possible to simulate the influence of vehicle circulation over the vector field.

The diffusion coefficient used was equivalent to the parameter considered in scenario three of [31], with the time given in weeks. In this configuration, the author simulates the dispersion of an atmospheric pollutant in the air. As the spore of the PP can be considered as a pollutant particle [9], the data had been used as a base.

For decay, an FS was constructed; whose input variables are the favorability map, the host map, and the fungicide application. The Sigma variable, which represents the decay coefficient (σ) of the DAE is, therefore, influenced by these three indicators.

The DAE, which modeled the dispersion of the fungus spores, is a second-order linear PDE, which has no analytical solution. Thus, a variational formulation was constructed to obtain approximations of its solution. For the spatial discretization, the Galerkin Finite Elements Method [32] was used, which uses the same class of functions for the base functions and test functions. For the temporal discretization, the Crank-Nicolson Method [33] consists of taking the middle instant between two subsequent moments. And to discretize the domain, the Finite Element Method [34] divides it into a finite number of simple subdomains, which in the case two-dimensional can be triangles, quadruples, among others.

### 3.3. Map of The Presence of Host Plants

This thematic map was based on the mapping provided by the National Supply Company (NASCO) for summer crops for the 2014/2015 harvest. The presence of host plants was mainly measured by the distance from the point to a property.

The mesh nodes located within the properties, or less than 50 m from the main highways, were considered the zero distance of hosts. Points outside the properties, but distant up to 1 km, were considered (to a greater or lesser degree) as close to hosts. All other nodes were considered distant from hosts, representing the absence of hosts.

### 3.4. Fuzzy Rust System

For ASR to occur, three conditions are indispensable: the existence of susceptible plants, virulent pathogens, and favorable environmental conditions [23]. Thus, it is clear which variables influence the occurrence of the disease, but not the function that relates to them. This uncertainty motivated the construction of the FS. As can be seen in Figure 1, it consists of three input variables (favorability, concentration, and host), and an output variable (fuzzy occurrence).

The variable Concentration refers to the number of spores per km^2^. The causal agent of ASR has a very short incubation period. In addition, a large number of spores per lesion are produced on the plant, which has high dispersion velocity [35]. For these reasons, the concentration was divided only between *High* (*H*), *Low* (*L*), and *Very Low* (*VL*).

The *Host* entry refers to the distance that the point is from a soybean plant. The greater the distance to the properties, the lower the chances of having a host plant on site. That is why a 1 km strip around the properties was established to assign different levels of belonging to the *Near* (*N*) and *Far* (*F*) subsets.

Upon reaching the soybean leaves, with at least 6 h of leaf wetness and favorable temperature, the spores germinate. If the conditions are favorable to infection and colonization, symptoms may arise in a few days [35]. Therefore, Favorability was defined as an input variable.

The fuzzy occurrence output represents the possibility of the disease occurring. The membership functions of this variable are *Very Low* (*VL*), *Low* (*L*), *Medium* (*M*), *High* (*H*), and *Very High* (*VH*). The closer to 1, the greater the chance of the disease happening. Figure 2 illustrates the reunion of these four variables. The rule base that establishes the relationship between them is composed of 11 rules and is described in Table 2.

For the resolution of the Fuzzy Rust System (FRS) the Mamdani inference method was used with defuzzification by the Center of Area Method. Thus, for each day of the analyzed period, and for each node of the mesh, a fuzzy occurrence value will be obtained, representing how susceptible the region is to the development of the disease.

## 4. Results and Discussion

The FRS was tested in the occurrences recorded for the disease between the 2015/2016 and 2019/2020 harvests. As the data that fed the model, directly or indirectly, are from 2018/2019, this crop is used as a reference. Maps of spore concentration, favorability, and host served as input variables for the FRS.

In Mato Grosso, the cities where the occurrences of the disease were recorded for the reference harvest are Sapezal, Jaciara, Tangará da Serra, Campo Novo do Parecis, Querência, Campo Verde, Comodoro, Gaúcha do Norte, Ipiranga do Norte, Campos de Júlio, Sorriso, Lucas do Rio Verde, Primavera do Leste, Nova Mutum, Feliz Natal, Itiquira, Rosário Oeste, and Sinop. That is why, for these places, the concentrations of fungus and favorability were identified for a specific period, based on the day of the ASR registration. Figure 3 shows where these cities are located in the state.

Some parameters used in the simulation of concentration varied in time and space, as was the case with transport and decay. The diffusion coefficient was based on the research of [31], and the values assigned to the parameters source and θ, in the searches of [36,37].

For instant zero, 31 October was considered, which is the day of the first occurrence of the disease recorded in the 2018/2019 harvest. In addition, for the other crops considered, these records took place from November or December. The concentrations were calculated until 30 April, when the soybean harvest for Mato Grosso ends, according to [38]. Figure 4 depicts the scattering of spores for the last day of each month.

Considering the localities where there was disease in the 2018/2019 harvest, the highest concentration recorded was 1,340,409 spores, in Lucas do Rio Verde. In addition, it is possible to observe that near the highways there is a higher concentration of spores. Thus, the problem of voluntary soybeans on the side of highways is evident.

It should be considered that the first signs of the disease take between 5 and 7 days to appear [35]. In addition, there may have been a delay in the observation of symptoms. Thus, the concentration data for these localities (where there was disease in the 2018/2019 harvest) were recorded for the interval between the ninth and fifth day prior to the occurrence record, in Table 3.

For the favorability map, it was necessary to interpolate the data obtained at the stations located in and around the state. The models tested were IDW at powers 1, 2, 3, 4, and 5, in addition to the geostatistical (kriging). Among the interpolators evaluated, by cross-validation, the IDW with power equal to two was the most representative. The geostatistical model occupied the third position, among the six interpolators analyzed. Thus, from the quadratic grade IDW, the thematic map was made (Figure 5), obtaining the favorability values for all mesh nodes, for January.

Reference [39] compared three estimation methods for soil pH (inverse of distance square, polynomial, and kriging) and concluded that kriging provided results with lower errors than the other two methods. For work in climatology, the IDW interpolator is used, because of its simplicity of handling, as in [40,41,42].

The interpolators showed good accuracy for estimating values in non-sampled locations for the ASR favorability variable. It is possible to observe that the highest rate of conditions favorable to the development of the disease occurs in the central region of the state, while the sites of lower favorability occur in the southern region.

For favorability data, those registered between the ninth, and fifth day prior to the occurrence record were also identified, in Table 4. The highest value among the five, that is, the highest favorability, was identified because it will be decisive for the attainment of fuzzy occurrence for each location.

The last entry of the FRS controller, which maps the pathogen hosts, is shown in Figure 6. In it are marked all points that are near the highways up to 50 m and all points that are in or far up to 1 km from the properties. All other nodes in the mesh were considered distant.

To verify the efficiency of the system, the occurrences recorded by the Antirust Consortium during the 2015/2016 to 2019/2020 harvests were raised, for the state of Mato Grosso. The locations and dates of the first records were taken for each city. Next, the fuzzy occurrence was examined for the 2018/2019 crop, which served to adjust the model, and for the other localities, where the disease was recorded in other harvests, to verify its percentage of correct answers.

For the day on which the highest favorability value occurred, the value of the concentration was identified. Thus, ordered pairs containing (concentration, favorability) were obtained for each occurrence. In these nodes, the host variable always assumed the value zero, since it is understood that there is a soybean plant at that point. Thus, for this ordered triplet (concentration, favorability, 0) were calculated fuzzy occurrences. The results are in Table 5.

The following comparisons are considered the data observed in Table 5. The possibility of the disease occurring was more than 50% for all locations. The lowest fuzzy occurrence value was 0.5 in Tangará da Serra and Feliz Natal, where small concentrations of spores and relatively low favorability were observed (in relation to the other records in the table). For favorability, the lowest recorded value was in Primavera do Leste. Although it was less than 0.5, the spore concentration ensured a fuzzy occurrence of 0.5318. Regarding concentration, the lowest value was observed in Itiquira, where favorability around 0.6 ensured that the chance of the disease happening was 53%. In the last two cases, values slightly higher than the minimum recorded were observed. For each day of occurrence recorded in Table 5, a map was generated. The 18 images were gathered in Figure 7, which follows the same sequence, from left to right, from top to bottom, of the cities in Table 5.

In this way, they were considered susceptible to the disease, localities that presented fuzzy occurrence greater than 0.5. Thus, we sought in the other harvests, which localities presented fuzzy occurrence greater than 0.5. For this, the interval for which the greatest favorability was sought varied between the ninth day before and the day after the occurrence because the data for this variable refers to the reference crop. Then, the procedure adopted for the occurrences recorded in 2018/2019 was repeated. In addition, it was observed for how many days during the harvest year, the localities presented fuzzy occurrence greater than or equal to 0.5. The data are summarized in Table 6.

Among all 84 first occurrences recorded during these five harvests, in only 16 the FRS could not correctly measure the susceptibility of the site to ASR. In this sense, the controller got his prediction right in 81% of the records. On the other hand, among the 38 cities where occurrences were recorded, only in three cities no value was recorded greater than 0.5 for any day of the crop year. By this analysis, the percentage of the hit was 92%.

In addition, Figure 7 and Table 6 show that the regions of Sapezal, Campo Novo do Parecis, Campo Verde, Ipiranga do Norte, Campos de Júlio, Sorriso, Lucas do Rio Verde, Nova Mutum, and Sinop are the most susceptible to the disease. Thus, for Mato Grosso, these are the areas that need more attention.

In view of what was presented, it is observed that, although the database was relatively small, its efficient use associated with the right tools made, it possible to build a robust model for the identification of more vulnerable regions in Mato Grosso. In addition, virulent strains of pathogens (fungus, bacteria, or viruses), hosts distributed in a region, and favorable environmental conditions are factors that, together, form the triangle of plant disease [43]. Thus, adapting the favorability data, the locality of hosts, and creating a DAE for the pathogen in question, the model presented in this research can serve as a basis for the study of numerous diseases that fit this profile.

## 5. Conclusions

For the development of Asian soybean rust, the presence of spores, the existence of favorable environmental conditions and host plants are determinants. Since the relationship between these variables (spores, environmental conditions, and hosts) is not known, we chose, in this research, to construct a Fuzzy System to relate them. To measure how favorable the environmental conditions are (favorability), the FS of [14] was reproduced, and the values were obtained in the existing weather stations in the state and in its surroundings. From there, the data were interpolated using the method that had the best result in cross-validation. The same procedure was adopted for the vector field of DAE used to estimate the presence of spores. For this variable, the values were calculated point by point, from numerical simulations. With the coordinates of the contours of the soybean-producing properties, the mesh points located within the properties were identified, as well as those near these properties; up to 1 km. Uncertainties were circumvented thanks to Fuzzy Logic, while partial differential equations solved the lack of data in relation to the fungus. The FRS was also dependent on these three variables, but obtaining the intake of spore concentration required greater dedication. Although DAE is widely used, there is not so much research dealing with air pollutants. Thus, many things that referred to the simulation of the data of this variable were innovative. In addition, a DES was also used to obtain one of the coefficients of the equation. The data that fed the FRS and the DAE come from 50 available points, from which it was possible to interpolate to another 2,712,265 points. Furthermore, taking into account that the correct answers were more than 80%, with this research, it is concluded that even a restricted database, when used efficiently, can return robust results. Although it has proven to be efficient for specifically estimating ASR, the controller may be useful for other cases. Given the generalist character of the tool suggested here, from adjustments, any disease could be modeled by the FS developed in this research. The constructed instrumental lends itself to efficiently simulating hypotheses indicated by the various specialists linked to the dispersion of the fungus causing ASR, establishing contingency strategies, and combating the disease, besides creating tooling for other possible diseases. It is also possible to mention the originality of transdisciplinary approaches in the effort to include several variables present and relevant to the judicious use of DAE in the modeling of analogous phenomena. Finally, it can be evidenced that this research is, in fact, a starting point for more actions of this type in problem situations, improving the methods and expanding their spectrum of effective use. In this way, this article can serve as a starting point for numerous studies, both for the modeling of other diseases and for their improvement. Therefore, it should be published.

## Figures and Tables

**Figure 1 sensors-22-00668-f001:**
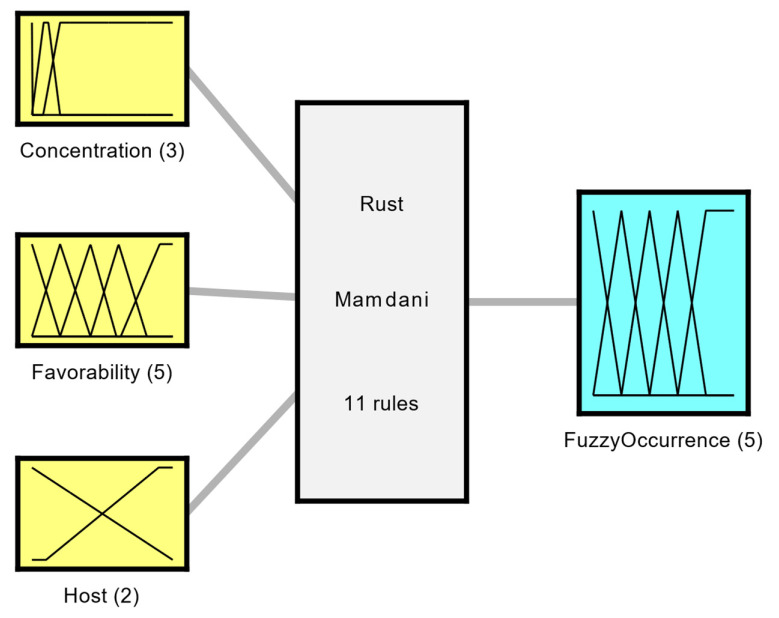
Organization chart with the input and output variables of the Fuzzy Rust System.

**Figure 2 sensors-22-00668-f002:**
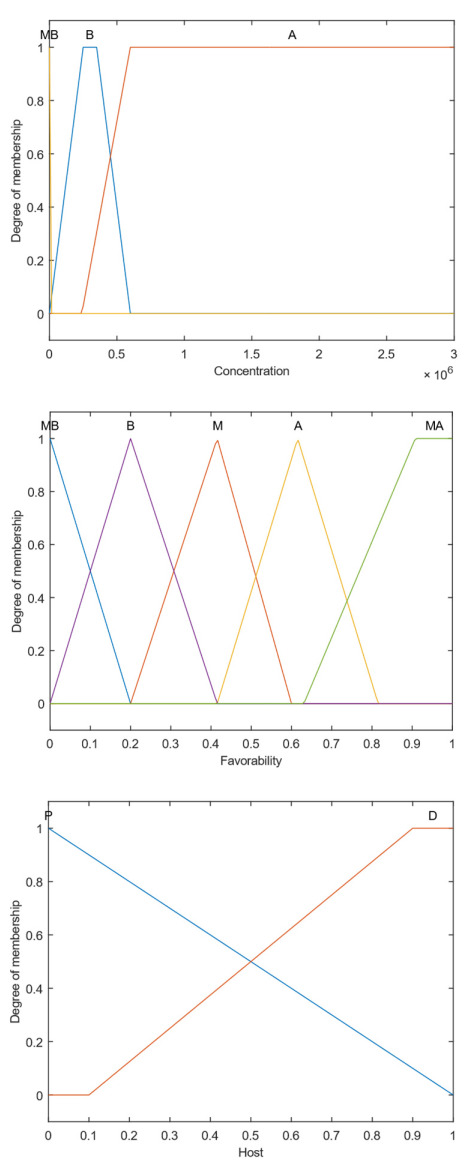
Fuzzy Sets representing the variables of Fuzzy Rust System.

**Figure 3 sensors-22-00668-f003:**
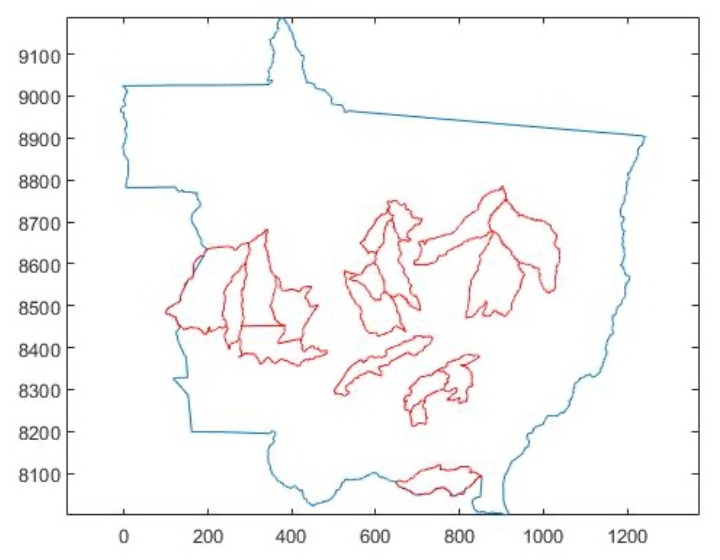
Localization of cities where occurrences of ASR were recorded for the 2018/2019 crop.

**Figure 4 sensors-22-00668-f004:**
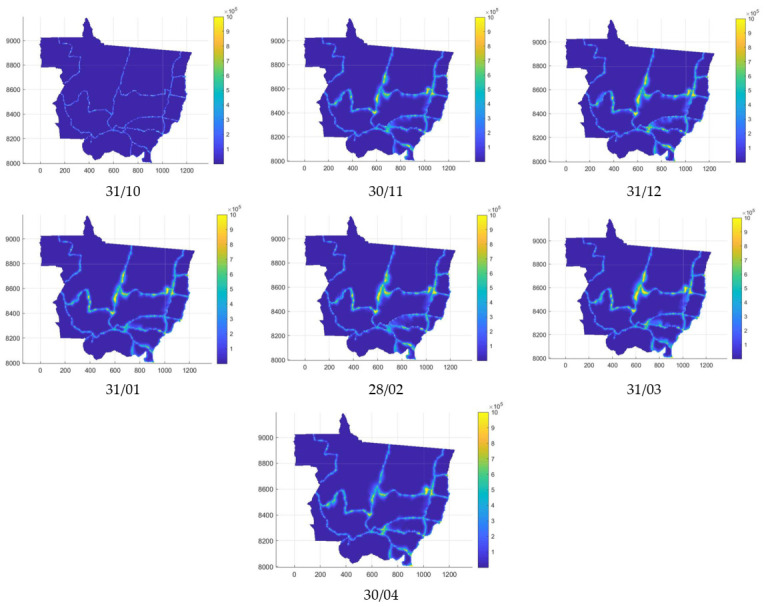
Spore concentrations per km^2^ for the last day of the month throughout the 2018/2019 harvest (October, November, December, January, February, March, April).

**Figure 5 sensors-22-00668-f005:**
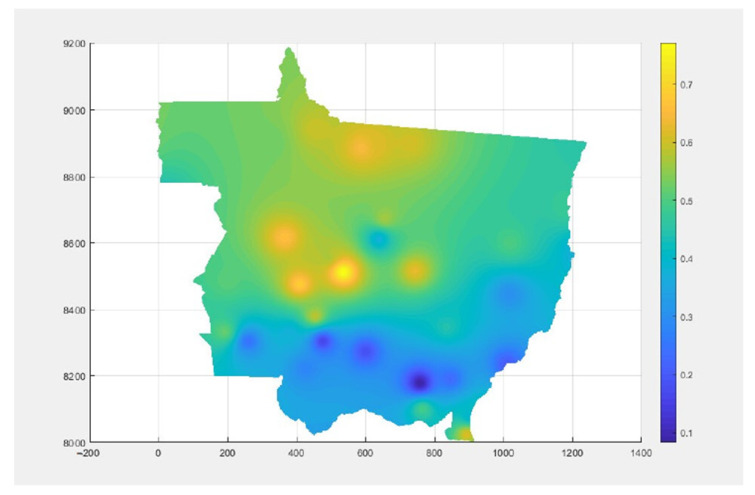
Map of favorability for the month of January 2019.

**Figure 6 sensors-22-00668-f006:**
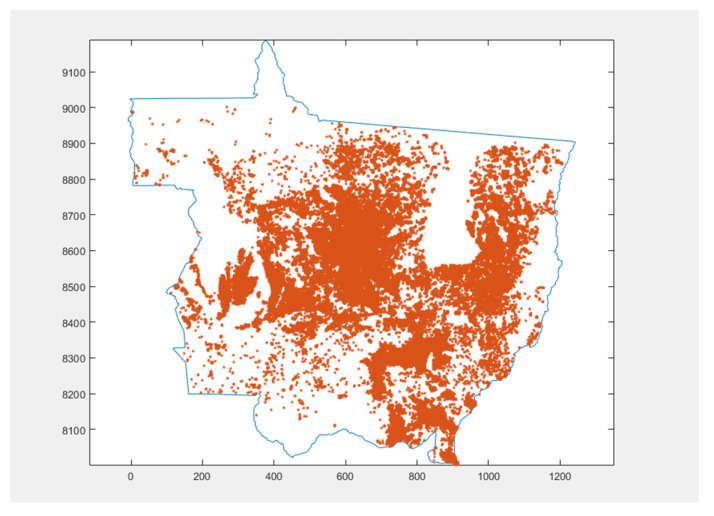
Map with the locations of fungus host plants.

**Figure 7 sensors-22-00668-f007:**
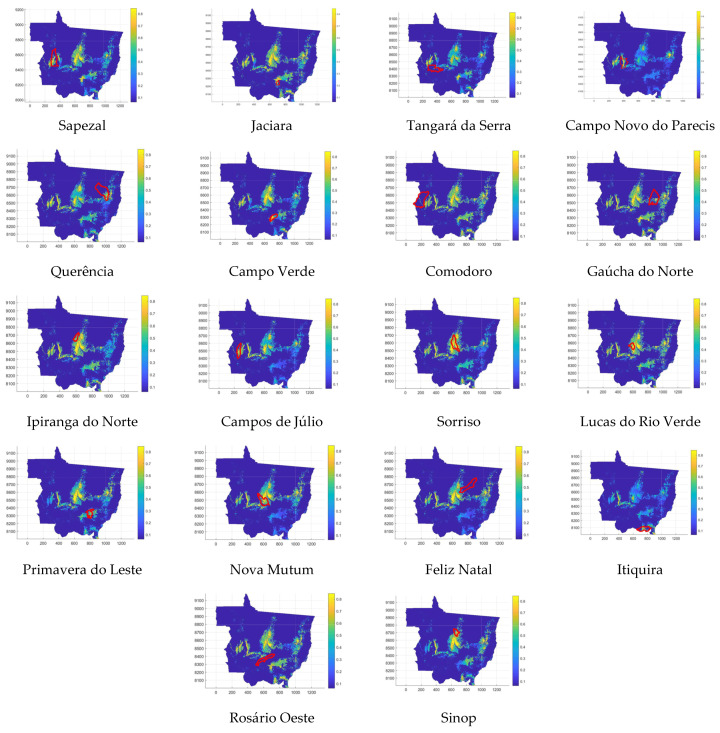
Fuzzy occurrence for localities (outlines in red) where occurrences were recorded in 2018/2019 harvest.

**Table 1 sensors-22-00668-t001:** Main related works.

Authors	Search Description
[10]	Built a decision tree for classifying ASR occurrences using meteorological variables as a basis. They modeled the influence of temperature and precipitation to predict situations of occurrence or non-occurrence of the disease.
[11]	Developed a neural network [12] to predict the severity of ASR for a given soybean cultivar. For this purpose, data from 73 epidemics and disease progress were used, which were measured weekly over two years for each experiment.
[13]	It produced a daily apparent infection rate model to simulate the severity of ASR. Combining meteorological variables and relevant biological criteria, they constructed a fuzzy logic system to estimate this rate that, applied to the differential equation of population dynamics, simulated the severity of the disease.
[14]	Studied the effects of leaf temperature and wetting on the monocyclic process of ASR in three-grain cultivars, from a fuzzy logic system and nonlinear regression models. With the implementation of the system, it was possible of observe favorable areas to the disease.
[15]	Developed empirical models to predict the short-range aerial movement of PP urediniospores using machine learning. These approaches were based on environmental variables and used urediniospore count data collected from active and passive spore traps in four environments in the southern United States. The study compared models using LASSO, Zero-inflated Poisson/regular Poisson regression, random forest, and neural networks.
[16]	Built an ASR aerobiology prediction system in order to evaluate the potential movement of pathogens from South America to the United States. The model is configured in a modular format to include all stages of the aerobiology process. To characterize the strength, and distribution of the spore source, colonization and disease progression in the affected sites, the submodel of host development, and disease progression, respectively, are conducted by climatic data. The system also provides for the release of spores, and canopy leakage in the areas of origin, mortality due to exposure to solar radiation during atmospheric transport, and wet deposition in the destination regions.
[17]	Artificial intelligence was used to increase the sustainability of the network for agricultural monitoring. Through fuzzy logic, it was possible to identify the most efficient places to be allocated sensor nodes, which detect local parameters, such as temperature, humidity, among others.
[18]	A monitoring technique based on the wireless sensor network has been developed. Sensor nodes are modeled through time-division multiple access, a technique of access to the middle to avoid the collision and control energy efficiently.
[19]	A comprehensive application of intelligent agriculture was proposed, aiming to obtain in large areas, productive harvesting of corn. It is about a system based on smart agriculture, Wireless Sensor Networks, and Drone communication.
[20]	Advanced fuzzy logic was used to develop a structure that allows controlling the commutation time of a pump in irrigation systems, according to the variables defined by the user. It is about an integrated agricultural monitoring system, using the sensor input.

**Table 2 sensors-22-00668-t002:** Fuzzy rule base for FRS.

Rule	Operator	Concentration	Favorability	Host	Fuzzy Occurrence
1	OR	VL	-	F	VL
2	AND	L	VL	N	VL
3	AND	L	L	N	L
4	AND	L	M	N	M
5	AND	L	H	N	H
6	AND	L	VH	N	H
7	AND	H	VL	N	L
8	AND	H	L	N	M
9	AND	H	M	N	H
10	AND	H	H	N	VH
11	AND	H	VH	N	VH

**Table 3 sensors-22-00668-t003:** The concentration of spores in locations where occurrences were recorded.

City	Register	9th Day	8th Day	7th Day	6th Day	5th Day
Sapezal	13/12/2018	104,535	77,176	99,957	143,572	167,451
Jaciara	18/12/2018	258,741	278,827	286,588	271,713	213,765
Tangará da Serra	26/12/2018	6111	5588	6036	5339	5248
Campo Novo do Parecis	27/12//2018	85,451	98,989	94,717	100,269	86,804
Querência	28/12/2018	526,776	619,361	697,381	677,027	709,100
Campo Verde	28/12/2018	705,336	638,635	572,333	537,763	574,251
Comodoro	03/01/2019	329,817	313,162	325,870	263,684	232,012
Gaúcha do Norte	08/01/2019	311,626	299,473	256,300	203,947	185,248
Ipiranga do Norte	10/01/2019	4283	4816	5553	6263	7117
Campos de Júlio	14/01/2019	208,944	208,391	196,693	190,338	188,681
Sorriso	15/01/2019	691,461	736,802	670,118	629,460	612,840
Lucas do Rio Verde	18/01/2019	1,261,478	1,279,944	1,306,017	1,194,618	1,227,913
Primavera do Leste	21/01/2019	273,590	270,741	365,522	278,794	250,307
Nova Mutum	22/012019	524,426	521,278	519,531	478,418	476,376
Feliz Natal	22/01/2019	2009	2403	2059	1722	1247
Itiquira	28/01/2019	1501	1615	1968	2013	2320
Rosário Oeste	22/02/2018	293,574	283,378	297,023	307,183	295,643
Sinop	15/03/2019	264,368	280,852	387,574	545,172	586,313

**Table 4 sensors-22-00668-t004:** Favorability data for ASR occurrence.

City	Register	9th Day	8th Day	7th Day	6th Day	5th Day
Sapezal	13/12/2018	0.5385	0.7017	**0.7279**	0.7049	0.6435
Jaciara	18/12/2018	**0.6310**	0.4671	0.3246	0.4840	0.4674
Tangará da Serra	26/12/2018	0.5788	**0.5819**	0.5385	0.0844	0.4466
Campo Novo do Parecis	27/12/2018	0.1291	0.1341	0.4940	0.4692	**0.7334**
Querência	28/12/2018	0.5423	0.5689	0.4821	0.5739	**0.6876**
Campo Verde	28/12/2018	**0.8123**	0.1958	0.2116	0.2138	0.4642
Comodoro	03/01/2019	0.8117	0.8259	0.8346	0.8462	**0.8534**
Gaúcha do Norte	08/01/2019	**0.6786**	0.6381	0.4597	0.3164	0.3900
Ipiranga do Norte	10/01/2019	0.6461	0.5223	0.5554	**0.7029**	0.6084
Campos de Júlio	14/01/2019	0.5778	0.6404	0.5778	**0.6650**	0.5754
Sorriso	15/01/2019	0.0792	**0.5831**	0.3307	0.1144	0.4966
Lucas do Rio Verde	18/01/2019	0.4178	0.5546	**0.6186**	0.3701	0.6028
Primavera do Leste	21/01/2019	0.3833	0.2249	**0.4546**	0.1760	0.1927
Nova Mutum	22/01/2019	**0.6275**	0.5895	0.4460	0.4033	0.4190
Feliz Natal	22/01/2019	**0.5817**	0.5797	0.3126	0.2459	0.2114
Itiquira	28/01/2019	**0.5994**	0.3961	0.1072	0.1186	0.2928
Rosário Oeste	22/02/2018	0.5292	0.3747	**0.6078**	0.5801	0.3808
Sinop	15/03/2019	**0.7288**	0.5676	0.5730	0.6264	0.5365

**Table 5 sensors-22-00668-t005:** Possibility of occurring ASR according to FRS.

City	Concentration	Favorability	Fuzzy Occurrence
Sapezal	99,958	0.7049	0.6000
Jaciara	258,741	0.6310	0.6157
Tangará da Serra	5588	0.5819	0.5000
Campo Novo Parecis	677,027	0.7334	0.8223
Querência	62,837	0.6876	0.600
Campo Verde	705,336	0.8123	0.8332
Comodoro	232,012	0.8534	0.6000
Gaúcha do Norte	311,626	0.6786	0.6582
Ipiranga do Norte	6263	0.7029	0.6000
Campos de Júlio	190,338	0.6650	0.6000
Sorriso	736,802	0.5831	0.8197
Lucas do Rio Verde	1,306,017	0.6186	0.8465
Primavera do Leste	365,522	0.4546	0.5318
Nova Mutum	524,426	0.6275	0.7802
Feliz Natal	2009	0.5817	0.5000
Itiquira	1501	0.5994	0.5324
Rosário Oeste	297,023	0.6078	0.6439
Sinop	264,368	0.7288	0.6287

**Table 6 sensors-22-00668-t006:** Records of fuzzy occurrence for all localities where there was at least one record of the disease over the five harvests and number of days that the locality was susceptible to the disease.

City	Harvest				Days
	2015/2016	2016/2017	2017/2018	2019/2020	
Pontes e Lacerda	NDR	NDR	0.6000	NDR	41
Poxoréu	0.0690	0.0987	NDR	0.0690	30
Cláudia	NDR	NDR	0.6000	0.6000	86
Confresa	NDR	0.7599	NDR	NDR	149
Pedra Preta	NDR	NDR	0.2420	NDR	49
Santo Antônio do Leste	NDR	0.6000	NDR	NDR	114
Rondonópolis	NDR	0.2479	NDR	NDR	51
Chapada dos Guimarães	NDR	0.6000	NDR	NDR	32
Canarana	0.6310	0.4927	NDR	NDR	60
Santo Afonso	NDR	NDR	0.5796	NDR	88
Marcelândia	NDR	NDR	NDR	0.0634	0
Paranatinga	NDR	NDR	NDR	0.6000	77
Diamantino	0.5745	0.5792	0.5792	NDR	68
Alto Taquari	NDR	NDR	0.6000	NDR	107
União do Sul	NDR	0.0633	NDR	NDR	0
Nova Ubiratã	NDR	NDR	0.6314	NDR	130
Tabaporã	0.0633	NDR	0.0633	0.0633	0
Alto Garças	0.8222	NDR	NDR	NDR	131
Vera	NDR	0.6017	0.6000	0.6000	94
Juscimeira	NDR	0.4362	NDR	NDR	57
Gaúcha do Norte	NDR	NDR	NDR	NDR	47
Campo Novo do Parecis	0.8222	0.7936	0.8351	0.8351	112
Feliz Natal	NDR	NDR	NDR	NDR	43
Sorriso	0.5719	0.8229	0.8398	0.8398	116
Comodoro	NDR	NDR	0.6683	NDR	137
Querência	0.6000	NDR	0.5870	NDR	117
Campos de Júlio	0.6000	NDR	0.6000	0.6000	109
Primavera do Leste	NDR	0.3492	0.2354	0.7324	70
Campo Verde	NDR	0.8332	0.8332	0.7241	116
Sinop	NDR	NDR	NDR	NDR	160
Itiquira	NDR	0.1641	NDR	0.1641	61
Jaciara	NDR	NDR	NDR	NDR	52
Rosário Oeste	NDR	0.6439	NDR	NDR	66
Ipiranga do Norte	NDR	NDR	NDR	NDR	84
Sapezal	0.6000	0.6000	0.6000	NDR	137
Nova Mutum	NDR	0.7615	0.7615	NDR	139
Tangará da Serra	0.6000	0.6000	0.6000	0.6000	122
Lucas do Rio Verde	0.7808	0.7471	0.8293	0.8457	161

Days—Number of days that the locality presented fuzzy occurrence greater than 0.5; NDR—No Disease Record.

## Data Availability

Not applicable.

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
