# Peer review of "Spatio-Temporal Modeling and Simulation of Asian Soybean Rust Based on Fuzzy System"

_sensors, 2022, doi:10.3390/s22020668_

Round 1

Reviewer 1 Report

In the paper, a fuzzy system is developed for gathering three variables as inputs, having as output the vulnerability of the region to the disease.

The paper is well organized.

I have some recommendations.

The aim, the motivation and the contriution of the paper should be emphasized.

Related works should be summarized in a table.

In Table 1, number of rules may be increased.

Some papers should be referenced as below:

Enhancing sensor network sustainability with fuzzy logic based node placement approach for agricultural monitoring, Computers and Electronics in Agriculture, 174, 1-10, 2020.

Energy-Efficient Technique for Monitoring of Agricultural Areas with Terrestrial Wireless Sensor Networks, Journal of Circuits, Systems and Computers, 29 (9), 1-17, 2020.

Smart agriculture with internet of things in cornfields, Computers & Electrical Engineering, vol.90, 2021.

Towards Smart Agriculture Monitoring Using Fuzzy Systems, IEEE Access, vol.9, 2021.

Author Response

We try to attend, as best as possible, all the reviewers' recommendations, considering the deadline established by the Editor (10 days). Changes implemented in the article are highlighted in yellow. The indicated numbering of the lines in the reply to the Reviewers refers to the original version.

Reviewer 2 Report

The paper deals with the regionally and globally important issue of spatial prediction of plant diseases. It uses a very up-to-date version of  the well proven Fuzzy-Logic approach to combine spatial information on the relevant factors determining the disposition of locations in the province of Matto Grosso for soybean rust. The procedure is straight foreward, the methodology is appllied in scientific way, the results are validated with independent data. So far the paper tries to present a relevant piece of science and should therefore eventually be pubished (after major revisions).

Now to the caveats:

0) the authors do not present any argument why they think the paper should be publishes in sensors

1) the English used in the paper is many times incorrect, which forces the reader to a lot of thinking about what the authors may have meant (see e.g. line 201: what is a "meteorological station of the automatic surfac"?, line 219: 50 seasons?, line 265 '...distance they are from the highway...', etc.). The paper needs serious language polishing!   

2) the figure captions are minimalistic. Figures captions together with the Fig. should be self-explanatory. In Fig.3 e.g. it is not clear what is shown (no unit, is it an time series, what is the exact time of each image?)

3) please provide a map with the location of the citis mentioned in table 2 and explain why cities are used as locations since one may assume that soybean rust is not occuring in cities because soybeans do not grow in cities.

4) in Fig.4 please explain why the IDW interpolation apparently shows remaining artefacts pointing at the locations of the stations, which were used for interpolation.  

5) in line 405: please explain what you mean by "For the day of greater favorability...."

6) in Fig.6.: please label each image for better readability, please explain the red polygons?

7) please properly format Table 5 and add caption!

8) in line 460: "They..." Who? In general, please reformulate he conclusions, it is not clear what you mean.

As ist the paper (not the science) lacks maturity and I urge the authors to spend the necessary seriousness on revising the paper so it can be published in Sensors.

Author Response

(The authors gave the same response as above.)

Round 2

Reviewer 2 Report

paper is fine as is.